# Mycosis Fungoides and Sézary Syndrome: An Integrative Review of the Pathophysiology, Molecular Drivers, and Targeted Therapy

**DOI:** 10.3390/cancers13081931

**Published:** 2021-04-16

**Authors:** Nuria García-Díaz, Miguel Ángel Piris, Pablo Luis Ortiz-Romero, José Pedro Vaqué

**Affiliations:** 1Molecular Biology Department, Universidad de Cantabria—Instituto de Investigación Marqués de Valdecilla, IDIVAL, 39011 Santander, Spain; garciadiaz.nuria@gmail.com; 2Department of Pathology, Fundación Jiménez Díaz, CIBERONC, 28040 Madrid, Spain; miguel.piris@quironsalud.es; 3Department of Dermatology, Hospital 12 de Octubre, Institute i+12, CIBERONC, Medical School, University Complutense, 28041 Madrid, Spain; pablo.ortiz@salud.madrid.org

**Keywords:** CTCL, mycosis fungoides, Sézary syndrome, diagnosis, molecular drivers, therapy

## Abstract

**Simple Summary:**

In the last few years, the field of cutaneous T-cell lymphomas has experienced major advances. In the context of an active translational and clinical research field, next-generation sequencing data have boosted our understanding of the main molecular mechanisms that govern the biology of these entities, thus enabling the development of novel tools for diagnosis and specific therapy. Here, we focus on mycosis fungoides and Sézary syndrome; we review essential aspects of their pathophysiology, provide a rational mechanistic interpretation of the genomic data, and discuss the current and upcoming therapies, including the potential crosstalk between genomic alterations and the microenvironment, offering opportunities for targeted therapies.

**Abstract:**

Primary cutaneous T-cell lymphomas (CTCLs) constitute a heterogeneous group of diseases that affect the skin. Mycosis fungoides (MF) and Sézary syndrome (SS) account for the majority of these lesions and have recently been the focus of extensive translational research. This review describes and discusses the main pathobiological manifestations of MF/SS, the molecular and clinical features currently used for diagnosis and staging, and the different therapies already approved or under development. Furthermore, we highlight and discuss the main findings illuminating key molecular mechanisms that can act as drivers for the development and progression of MF/SS. These seem to make up an orchestrated constellation of genomic and environmental alterations generated around deregulated T-cell receptor (TCR)/phospholipase C, gamma 1, (PLCG1) and Janus kinase/ signal transducer and activator of transcription (JAK/STAT) activities that do indeed provide us with novel opportunities for diagnosis and therapy.

## 1. Cutaneous T-Cell Lymphoma: Description of the Main Entities

Primary cutaneous T-cell lymphomas (CTCLs) vary in form from those of a very indolent type to extremely aggressive malignancies with bad prognosis. CTCLs are rare diseases with an annual incidence of around seven cases per million population [1]. The most common CTCLs are mycosis fungoides (MF) and Sézary syndrome (SS), which account for around 50% of all primary CTCLs [2,3]. This review focuses on MF/SS. The most recent classification of primary cutaneous lymphomas was recently published by the World Health Organization and the European Organization for Research and Treatment of Cancer (WHO-EORTC) [4,5].

### 1.1. Mycosis Fungoides

Mycosis fungoides (MF) is characterized by the proliferation of epidermotropic, cerebriform T cells. MF lesions present in the form of patches and plaques and may eventually develop tumors and leave the skin to involve peripheral blood, lymph nodes, and viscera.

MF most frequently appears in the sixth decade of life with a male/female ratio of 1.5–2:1 [2]. Its incidence has increased 300% from 1973 to 2008 (from 0.18 to 0.55/100,000), which exceeds that of all other lymphomas, estimated to be around 25% [4,6]. This rise in the incidence can be due to changes in the diagnostic criteria, population aging, and other unknown factors.

Typically, initial MF lesions are eczematous patches with well-defined borders and appear mostly on non-sun-exposed areas (swimsuit distribution). At this stage, diagnosis is difficult and can be delayed for 3–6 years [7]. This delay is also affected by the great clinical variability of MF lesions, such as poikilodermatous, hypopigmented, hyperpigmented, pustular, lichenoid, or bullous, to name but a few. Even MFs without clinical lesions have been reported [8].

Over years or decades, lesions can expand, become infiltrated, and progress to the plaque stage. Typically, islands of normal skin surrounded by lesional skin are frequent at this stage. Between 25% and 30% of patients can progress to tumoral stages with frequently ulcerated lesions that can appear on plaques or on previously normal skin. Some patients develop erythroderma (confluent erythema covering at least 80% of the total body surface). Mucosal involvement is rare and is usually associated with poor prognosis. Interestingly, the evolution of classic MF from patch or plaque to tumor and/or erythroderma develops in only 30% of cases [9].

Histopathologically, the features most commonly used in diagnosis include a combination of the following: (i) presence of band-like lymphocytic superficial infiltrate with atypical lymphoid cells within the epidermis, characteristically arranged along the basal layer, occasionally forming Pautrier’s microabscesses; (ii) lymphocytic atypia, mainly in the superficial component; (iii) abnormalities in the T-cell phenotype with loss of cluster of differentiation 7 (CD7), CD2, or CD5 markers, an abnormal CD4:CD8 ratio, or the intense expression of programmed death 1 (PD1) or forkhead box P3 (FOXP3) (CD8^+^ MF cases frequently show a marked epidermotropism); (iv) monoclonal T-cell receptor (TCR) γ and/or β rearrangement.

### 1.2. Mycosis Fungoides Variants

In the WHO-EORTC classification, some entities have been individualized because of their different prognosis.

#### 1.2.1. Folliculotropic MF

Folliculotropic MF (FMF) accounts for 5% of all primary cutaneous lymphomas. It is characterized by the presence of folliculotropic atypical infiltrates, frequently sparing the epidermis. Mucinous degeneration of follicles is frequent (follicular mucinosis). In some patients, atypical cells present tropism for eccrine sweat glands (syringotropism). Lesions usually appear on the head and neck area, the most typical being plaques with multiple follicular keratosis pilaris-like papules inside. Acneiform lesions, cysts, infiltrated plaques, or tumors can also appear. Alopecia is frequent and can be definitive for follicular destruction.

FMF came to be considered a different entity from classic MF because of its worse prognosis, with a 5 year survival of 70–80% [10]. However, two FMF variants have recently been described: “early FMF”, with a similar prognosis to that of classic MF, characterized by a 5 and 10 year overall survival (OS) of 92% and 72%, respectively, and “advanced FMF”, with a worse prognosis and an OS of 55% and 28% at 5 and 10 years, respectively [11]. Histopathologically, early FMF is typified by intrafollicular infiltrates and sparse or lichenoid perifollicular atypical lymphocytes. No cells outside the adventitial perifollicular dermis are present in early FMF. In advanced FMF, infiltrates are more extensive, involving a reticular, confluent, or diffuse dermis and containing medium-to-large tumor cells.

#### 1.2.2. Pagetoid Reticulosis

Pagetoid reticulosis (PR) is a rare entity (<1% of all primary cutaneous lymphomas) with an excellent 5 year survival of 100%. Clinically, PR presents with psoriasiform or hyperkeratotic plaques, usually solitary, on the extremities. Histopathologically, it shows a striking epidermotropism (“pagetoid”) of atypical cerebriform cells, mostly CD8^+^ and usually expressing CD30. T-cell rearrangement can usually be demonstrated [12].

#### 1.2.3. Granulomatous Slack Skin (GSS)

Granulomatous slack skin (GSS) is a very rare condition characterized by the development of pendulous folds that grow into large masses of lax skin, located mostly in the armpits, in the inguinal or submammary folds, or at the base of the neck. Its clinical course is usually indolent. Most cases are associated with MF. Hematoxylin and eosin (H&E) examination shows widespread granulomatous dermal infiltrate with atypical T cells, giant multinucleate cells, and elastophagocytosis. Cells are usually CD3^+^CD4^+^ [12].

### 1.3. Sézary Syndrome

Sézary syndrome (SS) accounts for 2–3% of all primary cutaneous lymphomas. It is characterized by the triad of erythroderma, generalized lymphadenopathy, and the presence of clonal atypical cerebriform T cells (Sézary cells) in the skin, lymph nodes, and peripheral blood [12,13].

Clinically, it mostly affects patients >60 years of age and has a bad prognosis (median survival of around 3 years). Strictly speaking, SS should be restricted to patients developing erythroderma de novo. The International Society of Cutaneous Lymphomas (ISCL) recommends that those patients with classic MF who develop erythroderma and peripheral blood involvement (stage B2 as detailed in Table 1) during evolution should be diagnosed as having “SS preceded by MF” or “secondary erythrodermic CTCL” [13]. In addition to erythroderma, patients present intense, frequently devastating pruritus, ectropium, palmoplantar keratoderma, and onychodystrophy. In addition, bone marrow is usually involved but infiltrates do not destroy its normal architecture. Furthermore, the course of this disease can be influenced by a variety of potential systemic problems and clinical preconditions, such as heart conditions in aged patients, hypoproteinemia, impairment of thermoregulation, and increased susceptibility to skin infections and sepsis. Historically, the most frequent cause of death in SS patients has been septicemia.

Histological features in SS are similar to those in mycosis fungoides, although the cellular infiltrates are frequently scarce, with slight or no epidermotropism [14].Neoplastic cells have a CD3^+^, CD4^+^, CD8^−^ phenotype, with loss of CD7 and strong uniform expression of PD1 [15]. Additionally, Sézary cells express cutaneous lymphocyte antigen (CLA) and the skin-homing receptor CCR4, as well as CCR7 [16,17].

### 1.4. Mycosis Fungoides/Sézary Syndrome Staging

MF/SS staging is based on TNM (for skin, lymph nodes and visceral involvement) completed with “B” (for blood involvement) (Table 1). Staging in the skin goes from T0 (no skin lesions present) to T4. T1 and T2 stages present patches or plaques involving less than or more than 10% of the total body surface, respectively. T3 indicates tumoral lesions, and T4 is used for erythroderma [19].

Lymph node (N) enlargement appears in 50% of cases (over 80% in the case of erythroderma). However, enlargement does not necessarily mean specific involvement by lymphoma. Lymph nodes are often reactive, enlarged as a consequence of the continuous entry of bacteria through the altered skin (dermatopathic lymphadenopathy). Core biopsy or lymph node excision is, therefore, recommended. Staging in lymph nodes goes from N0 to N3: N0 features no enlarged lymph nodes; N1 is dermatopathic lymphadenopathy and atypical lymphocytes should be absent, isolated, or in clusters of a maximum of 3–6 cells; N2 features aggregates of atypical cells but whose nodal architecture is preserved; N3 is characterized by partial or complete effacement of the nodal architecture [19].

Visceral involvement (M) is frequently found at autopsy (in around 70% of cases) but rarely in vivo [20].

Some patients develop peripheral blood involvement (B). The most recent staging proposed by EORTC ranges from B0 to B2: B0 cases have <250 atypical cells (CD4^+^CD7^−^ or CD4^+^CD26^−^), as identified by flow cytometry; B1 cases contain 250–1000 atypical cells; B2 features ≥1000 atypical cells plus clonality in peripheral blood [21]. In lymphopenic patients, “old” criteria should be used: B0 ≤5% Sézary cells; B2 should include peripheral blood clonality plus any of the following: (a) >1000 Sézary cells (determined by direct examination), (b) CD4:CD8 ratio >10 (as a consequence of increased CD4), (c) >40% CD4^+^CD7^−^, or (d) >30% CD4^+^CD26^−^ cells. B1 is anything intermediate between the B0 and B2 stages [19].

Most experts do not consider bone marrow involvement to be visceral involvement but instead peripheral blood involvement in the range of B2. In fact, bone marrow biopsy is not recommended unless special conditions, such as pancytopenia, occur.

Therefore, TNMB staging defines ten stages, from IA to IVB (Table 1).

### 1.5. Mycosis Fungoides/Sézary Syndrome Prognosis

Survival of MF is stage-dependent. Patients in stage IA do not seem to have shorter survival than the general population. Stages IA, IB, and IIA are considered “early MF” because of their prolonged OS (medians of 35.5, 21.5, and 15.8 years, respectively). Stages IIB and higher, including SS, are considered “advanced”, with median OS of <5 years (4.7 for IIB and IIIA, 3.4 for IIIB, 3.8 for IVA_1_, 2.1 for IVA_2_, and 1.4 for IVB) [22] (Table 1). 

Large-cell transformation (LCT) is the histopathological transformation of neoplastic small lymphocytes to a clonally identical, large-cell phenotype, which may occur in 20–55% of advanced MF cases [23]. Prospectively, the PROCLIPI (PROspective International Cutaneous Lymphoma Prognostic Index) study identified LCT, stage IV, age >60 years, and increased levels of lactate dehydrogenase as independent prognostic markers for shorter MF survival. Patients with 0–1 of these markers did not attain the median OS after >80 months, while those with 3–4 characteristics had a median OS of approximately 3 years [24].

## 2. A Landscape of Genomic Alterations in Mycosis Fungoides/Sézary Syndrome

Our knowledge of the molecular pathogenesis of MF/SS is incomplete. In the last few years, the use of next-generation sequencing (NGS) approaches revealed a rich landscape of genomic alterations potentially influencing the biology of these entities. From a mutational perspective, most studies have focused on performing whole-exon sequencing (WES) or targeted sequencing (TS), with more than 500 cases analyzed to date (Table 2). Overall, somatic single-nucleotide variants (SNVs) observed in MF/SS cases most frequently involve C > T transitions (40–74%) [25,26,27,28]. This mutational signature, which is much less common in other hematological cancers, is associated with exposure to ultraviolet (UV) light when occurring at NpCpG sites. This is currently considered a major driver of some tumors affecting the skin, such as cutaneous melanoma or Merkel cell carcinoma [29,30,31]; however, remarkably, no correlation has been noted between the presence of this mutational signature and history of therapeutic UV treatment in MF/SS lesions (localized radiotherapy or extracorporeal photopheresis) [26,27].

Functionally, genomic alterations in MF/SS cases, including somatic mutations (^mut^), amplifications (^amp^), or deletions (^del^), are frequently detected in genes that are well-known participants in key cellular activities like DNA damage (e.g., *TP53*^mut,del^ and *ATM*^mut,del^), TCR signaling (*PLCG1*^mut^*, NFATC2*^mut^*, NFAT5*^mut^*, ZEB1*^del^, and *PRKCQ*^amp,mut^), NF-κB signaling (*TNFRSF1B*^mut^ and *CARD11*^mut,amp^), CCR4/MAPK signaling (*CCR4*^mut^*, NRAS*^mut^*, KRAS*^mut^, and *MAP2K1*^mut^), JAK/STAT signaling (*JAK1*^mut^*, JAK2*^amp^*, JAK3*^mut^, *STAT3*^amp^, and *STAT5B*^mut,amp^), cell migration (*RHOA*^mut^*, VAV1*^mut^, and *PREX2*^mut^), and chromatin architecture (*CTCF*^mut,del^, *ARID1A*^mut,del^, and *TRRAP*^mut,amp^) [25,26,27,28,32,33,34,35,36,37,38,39,40,41,42,43,44] (Table 2).

Among these, highly prevalent chromosomal deletions have been described in tumor suppressor genes, such as *TP53, ZEB1, RB1, PTEN, DNMT3A*, and *CDKN1B*. Interestingly, the ratio of *TP53* deletions to mutations is significantly higher in MF/SS than in other tumors (5.1 vs. 1) [27].

Gene expression profiling studies have shown that increased signaling from the TCR can be considered a MF/SS driving force, together with signaling derived from TNF [26,51]. In addition, recent RNA-seq analyses (>300 cases) have also helped identify relevant genes, pathways, and specific transcriptome signatures that are specifically deregulated in MF lesions and SS cells [26,28,33,41,45,46,47,48,49,50]. On the one hand, lessons learned from these contributions highlight the rich molecular inter- and intra-tumoral heterogeneity in MF/SS cases. Moreover, single-cell analyses revealed individual molecular profiles among different patients and even between tumoral cells from the same patient. Despite this heterogeneity, some genes participating in the control of the cell cycle, cellular proliferation, and survival have been found to be commonly deregulated in highly proliferating malignant T cells [49]. The cellular activities carried out by genes with deregulated expression in MF/SS are mostly redundant in conjunction with those associated with the mutated genes.

On the other hand, TCRβ and γ NGS studies have fueled considerable controversy regarding the degree of clonal heterogeneity present in MF/SS cases. Strikingly, intra-tumoral heterogeneity was recently explained by Iyer et al., who obtained a median of six subclones with a branched phylogenetic relationship [43]. Furthermore, a model has been proposed in which the initial malignant transformation in MF occurs in T-cell precursor cells, rather than in mature memory T cells, before TCRβ and TCRα rearrangements arise [50]. These findings contrast with those published by de Masson and coworkers, whereby the tumor clone frequency (TCF > 25%) in affected skin, measured by NGS of the *TCRB* gene, was an independent prognostic factor of both progression-free survival (PFS) and OS in patients with an early MF/SS stage [52].

## 3. A Malignant Network of Signaling Mechanisms Drives Mycosis Fungoides/Sézary Syndrome

From a molecular perspective, the pathogenesis and progression of MF/SS can be driven by an intricate network of malignant mechanisms, highly influenced by deregulated TCR/PLCγ1–NFAT, TNFR–NF-κB, and JAK–STAT signaling pathways, triggered by (i) aberrant autocrine or paracrine stimulation of T-cell receptors (such as CCR4, TCR, TNFR, or interleukin receptors) by cytokines, interleukins, or growth factors generated by the transformed T cells and/or by the tumoral microenvironment, and (ii) the acquisition of multiple genetic alterations resulting in mutated receptors and/or their different intracellular effectors and leading to the activation of the transcription factors NFAT, AP-1, NF-κB, and STATs (Figure 1).

### 3.1. TCR/PLCγ1 Signaling

T-cell receptors (TCRs) are protein complexes composed of six different polypeptides. The TCRs of most T cells comprise TCRα, TCRβ, CD3γ, CD3δ, CD3ε, and CD3ζ, but some T cells, mostly located in the mucosal compartments, carry γδ TCRs [53]. TCRαβ subunits have immunoglobulin-like variable domains that recognize peptide antigens associated with major histocompatibility complex (MHC) molecules expressed on the surface of antigen-presenting cells (APCs). TCRαβ are associated with the invariable subunits of the CD3 complex (γ, δ, ε, and ζ) that enable signal transduction. When T cells interact with APCs, the TCR complex is assembled and triggers the formation of immunological synapse (IS), which controls T-cell activation, as well as helper and cytotoxic effector functions. TCR–CD3 complexes can recruit the Src family protein tyrosine kinases, such as LCK [54]. Amongst its activities in T cells, LCK maintains the signaling required for the survival of naïve T cells and phosphorylates Zeta-chain-associated protein kinase 70 (ZAP-70), which activates the linker for activation of T cells (LAT), an adaptor protein that assembles multiple proteins, such as phospholipase C gamma 1 (PLCγ1) [55,56]. In T-cell lymphomas, TCR downstream signaling can be triggered by TCR–MHC interactions within the tumor microenvironment (e.g., dendritic cells and macrophages) [57] and/or mutated co-stimulatory receptors (e.g., CD28) and effectors (e.g., PLCγ1), leading to malignant growth, survival, deregulated cytokine production, altered T-cell phenotypes, and dysfunctional formation of the IS [58].

PLCγ1 is an enzyme that catalyzes the formation of inositol 1,4,5-trisphosphate (IP_3_) and diacylglycerol (DAG) from phosphatidylinositol 4,5-bisphosphate (PIP_2_). On one hand, IP_3_ triggers an acute and transient Ca^+2^ release from the endoplasmic reticulum into the cytoplasm, where it activates its cognate targets, including calmodulin (CaM). In turn, this activates calcineurin (CaN) signaling toward the dephosphorylation and activation of NFAT transcription factors. On the other hand, DAG is a physiological activator of conventional and novel PKCs (α, β, γ and δ, ε, η, θ, respectively).

*PLCG1* (PLCγ1 gene) is a recurrently mutated gene in MF/SS. The average percentage of cases harboring mutations in *PLCG1* is 11%, ranging from 5.5% to 21% of the cases [26,27,28,32,34,35,37,38,43]. The most frequent *PLCG1* mutation in MF/SS cases is S345F, which is located in the catalytic PLC domain. This mutant, along with others such as S520F or the VYEEDM1161V indel, have been shown to promote the activation of downstream nuclear effectors such as NFAT, AP-1, and NF-κB [32,59]. Moreover, using a combination of mutational and immunohistochemistry (IHC) analyses of MF/SS cases, mutated *PLCG1* (mostly S345F) correlated with positive nuclear staining of activated NFAT [32]. In addition, genomic alterations in *NFAT5* have been detected in up to 7% of MF/SS cases, and amplification and overexpression of *JUNB* (a member of the AP-1 transcription factor complex) were identified in a high proportion of the cases analyzed [60,61]. Highly significant biological activity in T cells is mechanistically attributable to PLCγ1-mediated transcriptional activation of interleukin-2 (IL-2) via NFAT and AP-1 in T-cells [62]. Taking into account the high percentage of cases with genomic alterations in members of this pathway and its associated activating mechanisms, it is possible to conceive that deregulated PLCγ1 can control this intricate signaling network and determine the biological behavior of MF/SS lesions.

### 3.2. The PKC/NF-κB Axis

Downstream of PLCγ1, mutations, but mostly amplifications in *PRKCB* (2.5%; protein kinase C, isoform β, PKCβ) and *PRKCQ* (30%; PKCθ), have been detected in MF/SS [27,38,40]. *PRKCQ* amplifications do not overlap with *PLCG1* mutations, which strongly suggests that both participate in the same signaling axis. Therefore, at least 50% of the MF/SS cases harbor genetic alterations in PLCγ1/PKCβ–PKCθ downstream signaling. PKCβ and PKCθ belong to a family of serine/threonine kinases that are activated by DAG and phorbol esters. PKCθ is expressed in skeletal muscle and lymphoid organs, predominantly in the thymus and lymph nodes. Among hematopoietic cells, PKCθ is most abundant in T cells and is expressed at much lower levels or is even undetectable in B-cells, macrophages, neutrophils, and erythrocytes [63,64]. PKCθ has unique properties that distinguish it from other T-cell-expressed PKCs: first, it can translocate to the IS upon TCR-MHC assembly; second, these translocations can be selectively regulated by the VAV/RAC pathway via the actin cytoskeleton. Activated PKCθ positively regulates NFAT, AP-1, and NF-κB transcription factors leading to T-cell activation, proliferation, survival, and differentiation processes [65,66,67,68].

NF-κB transcription factors consist of two dimers composed of the p50–p65/RELA (canonical) and p52/RELB (non-canonical) subunits. Canonical NF-κB (p50–p65/RELA) is a major target of PKCθ activated via CARD11 in T cells [69,70]. The complex consisting of CARD11, BCL10, and MALT1 promotes the phosphorylation and activation of the inhibitor κB kinases (IKK) complex, which, in turn, phosphorylates IκB (an NF-κB repressor), leading to its ubiquitination and proteosomal degradation. In the non-canonical pathway, the NF-κB-inducing kinase directly phosphorylates and activates IKKα, which phosphorylates p100, inducing its proteasomal processing into p52. Then, p52 heterodimerizes with RelB and translocates to the nucleus (reviewed in [71]). In MF/SS, recurrent alterations in *CARD11* and in non-canonical NF-κB genes (*NFKB2* and *RELB*) have been described (Table 1 and [26,27,28,32,34,35,37,38,43]). As an alternative to PKCs, NF-κB can be activated downstream of membrane receptors that respond to extracellular stimuli such as TNFR and TLR/IL1R [72,73]. Downstream of these, and in MF/SS lesions, signaling mediators such as TRAF proteins and kinases such as TAK1, can trigger NF-κB activation and control malignant T-cell responses and important biological activities, including differentiation toward Th1, Th2, and Th17 cell lineages [71,74]. To do so, NF-κB regulates transcription of essential T-cell genes that code for cytokines (e.g., *IL4, IL6*, and *TNF*), cell-cycle regulators (*MYC* and *CCND1*)*,* survival proteins (*BCL2*), and angiogenesis mediators (*IL8* and *VEGF*), to name a few examples [73,75].

### 3.3. JAK/STAT Signaling

The JAK family of cytoplasmic proteins consists of four tyrosine kinase members (JAK1, JAK2, JAK3, and Tyk2) that form heterodimers and homodimers in the cell membrane, transducing signals toward STAT proteins through several transmembrane receptor families. JAK proteins comprise four domains: the FERM and SH2 domains, which are essential for the interaction with the receptors, and the pseudokinase domain, which interacts with the kinase domain to regulate the latter’s catalytic activity [76].

Seven STAT proteins have been identified in human cells: STAT1, STAT2, STAT3, STAT4, STAT5A, STAT5B, and STAT6. The functional domains of STATs include a coiled domain that is necessary to ensure interaction with other proteins, a DNA-binding domain, an SH2 domain, which mediates binding to phosphorylated tyrosine residues of other STATs (dimerization), and a C-terminal transactivation domain required for full transcriptional activation [77].

Upon activation of the receptor, JAKs can recruit members of the STAT family and phosphorylate them in tyrosine residues. Upon phosphorylation, STATs form dimers, which translocate to the nucleus and act as transcription factors inducing gene expression of factors involved in apoptosis/survival, angiogenesis, proliferation, and differentiation via the DNA-binding domain [78]. In some cases, signaling through STATs can be triggered by receptors with intrinsic tyrosine kinase activity, such as epidermal growth factor receptor (EGFR) and fibroblast growth factor receptors (FGFR1–3), or by cytoplasmic tyrosine or serine/threonine kinases, such as those of the Src family (e.g., SRC, FYN, or LCK) or PKCs [79].

The “canonical” signaling phosphorylation cascade at tyrosine residues (e.g., Y701 in STAT1, Y705 in STAT3, Y694 in STAT5A, and Y699 in STAT5B) [80] and subsequent SH2 domain-mediated homodimerization or heterodimerization of STATs have customarily been described as the essential requisite for biological activity of the proteins. However, phosphorylation in serine residues, unphosphorylated STATs, and other chemical modifications also have important roles in controlling STAT-mediated transcription [81,82,83].

Deregulation of the JAK/STAT pathway has been described in many hematological malignancies, where, presumably, it confers a selective advantage driving transformation into a malignant precursor cell. Most JAK mutations found in T-cell malignancies can be detected in *JAK1* and *JAK3*, in contrast to *JAK2* mutations, which are mainly found in myelodysplastic syndromes. It is also well known that aberrant activation of JAK1/3–STAT3/5 is an important feature of T-cell lymphomas including MF/SS [84]. Providing further evidence of their involvement in CTCL, highly recurrent genetic alterations (SNVs and copy number variants) have been detected in JAK/STAT genes in ≥60% of the MF/SS cases analyzed [25,27,32,35,36,37,38,40,41,43]. Notably, STAT3 and STAT5B are amplified in 60% of patients. Mutated STAT proteins have also been identified, although to a lesser extent, and are predominantly localized in the highly conserved SH2 domain [85]. Lastly, activating mutations in JAK proteins have also been identified in MF/SS in 4% of cases.

### 3.4. STAT3 Activation in Mycosis Fungoides/Sézary Syndrome

A high percentage of MF/SS cases display nuclear accumulation of phosphorylated STAT proteins as detected by IHC. In the context of the malignant network of MF/SS signaling mechanisms described above and in Figure 1, a comparative IHC study was recently performed in a cohort of 78 MF cases at early and advanced stages. It focused on analyzing the nuclear expression of NFAT, NF-κB (canonical and non-canonical proteins) and P-STAT1, P-STAT3, and P-STAT5. No significant correlation was observed between any MF stage and positive nuclear expression of NFAT, NFκB, P-STAT1, and P-STAT5; however, remarkably, positive P-STAT3 was significantly correlated with advanced stages [42]. In addition, P-STAT3 has also been reported in tumor stage CTCLs [86,87,88]. Therefore, activated STAT3 can constitute a mechanistic driver with relevant implications for tumorigenesis and progression of MF/SS cases.

Taking into account the imbalanced percentages observed between MF/SS cases with mutated JAK genes (about 4%) and those with activated STAT protein, it is reasonable to surmise that their malignant activities may also be influenced by alternative mechanisms that could include (i) mutations and/or deregulated activity of alternative cytoplasmic kinases to JAKs and/or members of the malignant signaling network controlled by PLCγ1, (ii) a proinflammatory microenvironment generated by the malignant T cells and other non-transformed reactive cells attracted to the lesions, and/or (iii) immune responses to bacterial colonization in the compromised skin barriers of MF/SS lesions. In this regard, increased prevalence of *Staphylococcus aureus* has been detected in MF/SS patients, which can contribute to disease flares by, for example, inducing activation of STAT3 and expression of STAT3-regulated cytokines in the malignant T cells or by stimulating FOXP3 expression in malignant T-cells, whose expression and function can predict early tumor development. [89,90,91]. A recent study linked methicillin-resistant *S. aureus* with erythrodermic CTCL patients, highlighting the relevance of such infections in the pathogenesis of this disease [92,93].

### 3.5. CCR4/CCR7 Signaling

C–C chemokine receptor 4 (CCR4) is the receptor for C–C motif chemokine ligands 17 and 22 (CCL17 and CCL22) and mediates the migration of Th2 cells and regulatory T cells to the skin [94]. CCR7 is the receptor for CCL19 and CCL21 and is required for the homing of T cells to the lymph nodes [95]. Both receptors are G protein-coupled receptors (GPCRs) containing seven transmembrane domains that mediate their downstream signals through heterotrimeric G proteins. Upon the stimulation of its ligands, CCR4 and CCR7 can trigger activation of downstream effectors, such as MAPK–ERK, RhoA and Rho-associated protein kinase (ROCK) through β-arrestin2-dependent mechanisms [96,97,98,99]. Downstream of these, CCR4/7 can thereby control transcription mediated by AP-1 and NF-κB, as well as regulate cytoskeletal dynamics and cell migration (Figure 1). Furthermore, Th2 cell migration is also mediated by PLCγ1 signaling downstream of CCR4/7 activity [100,101]. CCR4 expression has been detected in both MF and SS, whereas CCR7 and L-selectin are mostly expressed in the leukemic variants of CTCL, such as SS.

## 4. Therapy for Mycosis Fungoides/Sézary Syndrome

Currently, there is no curative treatment for advanced MF/SS, with the possible exception of allogenic stem cell transplantation and radiotherapy in unilesional MF. To date, no treatment has been shown to increase survival significantly, although it is difficult to design clinical trials with cure or survival as clinical endpoints when treating a disease, such as MF, in which survival is already very prolonged. A more realistic approach might be to search for prolonged responses and increased patient quality of life with minimal adverse events during treatment.

Various guidelines have been published by the National Comprehensive Cancer Network (https://www.nccn.org/, accessed on 1 December 2020), EORTC, and ESMO, among others [102,103]. All concur that the quality of the evidence generated is hampered by the small number of randomized, well-controlled clinical trials performed in MF/SS. A stepwise, stage-adapted approach is, therefore, recommended for the treatment of these patients, considering age, patient general status, extent of lesions, rate of disease progression, and previous therapies [102]. In general, skin-directed therapies are used to treat early stages of MF and FMF as a first-line treatment, while systemic therapies, usually combined with skin-directed treatment, are more commonly used to treat more advanced stages. The treatment for FMF can be similar to that for classic MF, but the time to response and treatment duration in early FMF is more prolonged [104]. With regard to the other MF variants, PR and GSS, treatment with radiotherapy or surgical excision has been recommended [12,105,106].

### 4.1. Skin-Directed Therapies

#### 4.1.1. Topical Corticosteroids

Corticosteroids impair not only lymphocyte binding to the endothelium, but also intercellular adhesion. They are widely used as a palliative treatment for individual lesions in the early patch/plaque stage. In a single study of 79 patients with stage IA/IB, the twice-daily use of high-potency topical corticosteroids (e.g., clobetasol propionate) showed an overall response rate (ORR) of 94% for stage IA and 82% for stage IB, with complete response (CR) rates of 63% and 25%, respectively [107]. High-potency topical corticosteroids are usually used in daily practice. Responses are frequent but short-lived.

#### 4.1.2. Topical Mechlorethamine

Mechlorethamine, also known as nitrogen mustard (NM), is a cytotoxic chemotherapy agent approved by the United States Food and Drug Administration (FDA) for the treatment of MF stage IA/IB in patients who have previously received skin-directed therapy. The 0.02% gel preparation gave response rates of 58.5% with 13.8% CR [108]. It was recently authorized in Europe as a first-line treatment for early MF stages (https://www.ema.europa.eu/en/documents/product-information/ledaga-epar-product-information_en.pdf, accessed on 1 December 2020).

#### 4.1.3. Ultraviolet Phototherapy

A consensus guideline for the use of phototherapy in MF/SS, for clinical practice and clinical trials, was recently published by the United States (US) Cutaneous Lymphoma Consortium [109]. 8-Methoxypsoralen, supplied orally, plus ultraviolet A (PUVA), and narrowband ultraviolet B (nbUVB) are the most widely available phototherapy options. nbUVB is recommended for early MF/SS stages that are characterized by patches only. PUVA is the phototherapy recommended for plaque disease and for patients with dark skins. The CR of nbUVB ranges from 54% to 90% of patients (average: 84%), depending on the study and the different ways in which disease clearance is defined. CR rates for PUVA are 85% for stage IA, 65% for stage IB, and 85% for stage IIA [109]. If the patient has an insufficient response or immediate relapses, phototherapy can be combined with systemic options such as retinoids, rexinoids (bexarotene), or interferon α.

In general, patients treated with phototherapy can develop erythema and pruritus, but the most important side-effect is the development of secondary skin cancer. Although some large studies addressed the potential risk of acquiring skin cancer after phototherapy treatment in patients with psoriasis and other skin disorders [110], these studies in MF/SS patients are limited. In a follow-up study of early-stage MF patients treated with PUVA as continuous maintenance about every 6 weeks, 26% developed non-melanoma skin cancer, including squamous-cell carcinoma and basal-cell carcinoma [111]. The long-term follow-up of patients or alternative treatment should be considered in order to prevent and/or treat possible adverse effects of phototherapy in MF patients.

#### 4.1.4. Total Skin Electron Beam Therapy

Total skin electron beam (TSEB) therapy is used when the disease is distributed along the entire skin surface. Electrons are generated in a linear accelerator and attenuated to penetrate the skin to a limited depth, thereby limiting its potential toxicity to internal organs. TSEB is recommended for first-line treatment of MF from stage IIB to stage IVB and as a second-line treatment for stages IA, IB, and IIA. Recently, reduced radiation doses (12 Gy) were successful compared with higher doses, with an ORR of 87.5% and lower toxicity [112]. Progression-free survival is higher for patients in stage IB than stage IIB (26.5 vs. 11.3 months) [113].

#### 4.1.5. Localized Radiotherapy

Localized and superficial radiotherapy (RT) provides effective palliative treatment for individual lesions and induces long-term remission in unilesional disease [114]. Photons and electrons can both be used. Responses are very high, around 90% of treated lesions, but the treatment is not curative. It can be used alone or as an adjuvant treatment in combination with other skin-directed or systemic therapies for patients with early-stage MF.

### 4.2. Systemic Therapies

#### 4.2.1. Retinoids

Retinoids are immunomodulating agent derivatives of vitamin A whose function is to interact with nuclear receptors (both retinoic acid (RAR) and retinoic X (RXR) receptors). Responses of acitretin and etretinate, both targeting RARs, have been shown in 44–67% of patients [115,116]. Bexarotene is an inductor of apoptosis, which inhibits metastasis and angiogenesis, and it is the only retinoid specifically developed and approved for the treatment of refractory, advanced-stage CTCL. It has shown an ORR of 45–54%, although it is commonly used in combination or as maintenance therapy [117]. The most popular combination is with PUVA [118]. This has a UVA-sparing effect, reducing the risk of the long-term problems of ultraviolet irradiation. Other combinations with extracorporeal photopheresis or interferon are frequently used.

#### 4.2.2. Interferon α

Interferons (IFN) are polypeptides with antiviral, cytostatic, and immunomodulating functions. Of these, IFN-α is the one approved for treating MF/SS. Various studies have reported an ORR of up to 88% in stages IB/IIA and 63% in stages III or IV, with CRs in 10–27% of cases [119]. Similarly to bexarotene, IFN is frequently used in combination with PUVA, increasing responses and sparing UVA [120]. Combination with extracorporeal photopheresis is also common.

PEGylated IFN-α (PEG-IFNα) has also been used off-label to treat MF with an ORR of 83% [121]. Weekly injections are more acceptable to patients, since classic IFN requires injections three times a week, and adverse event frequency seems to be lower, especially for flu-like symptoms. Currently, PEG-IFNα remains the only available option.

#### 4.2.3. Extracorporeal Photopheresis

Extracorporeal photopheresis (ECP) consists of white-cell separation with apheresis, mixing the buffer with the photosensitizer drug 8-methoxypsoralen with posterior UVA irradiation. UVA-irradiated white cells are reinfused. ECP is indicated in particular for erythrodermic MF and SS patients. Recently, updated guidelines for the use of ECP were published [122]. It has an excellent safety profile and very rare adverse events, with an ORR between 31% and 86% and CRs ranging between 0% and 62%.

Combination treatment options with, for example, bexarotene or interferon, are frequently used. Maintenance treatment is usual after response [123].

#### 4.2.4. Chemotherapy

Single-agent and combination chemotherapy regimens have been used to treat non-Hodgkin lymphomas since the 1970s. In general, MF lesions are chemoresistant because their proliferation rate is low. In early MF, chemotherapeutic agents are not generally recommended, the exception being low-dose methotrexate, which is acceptable. In this regard, it is well known that aggressive treatment of early MF can induce a higher percentage of responses, but the survival curves are similar to those of skin-directed therapy [124]. On the other hand, chemotherapy can be used on MF stages IIB and higher. Low-dose methotrexate (5–30 mg weekly) has been used to treat plaque stage or erythrodermic MF, with ORRs of 33% and 58%, respectively [125].

A review of single chemotherapeutic agents including 526 patients reported ORRs of 66% (median duration 3–22 months) with 33% CRs [126].

Currently, gemcitabine and pegylated liposomal doxorubicin are the most frequently used monotherapy agents. An EORTC clinical trial with pegylated liposomial doxorubicin reported an ORR of 40.6% with a 6 month response duration [127].

Other agents, such as pralatrexate, have been approved by the FDA, but not by the European Medicines Agency (EMA), for the treatment of peripheral T-cell lymphomas including MF. However, its high toxicity hampers its generalized use [128].

Lastly, oral chlorambucil has been used in combination with prednisone to treat SS. Initial reports claimed twice the survival time of alternative therapeutic regimens, although its long-term use can provoke myelosuppression and the eventual risk of leukemogenesis [129].

#### 4.2.5. Allogenic Stem-Cell Transplantation

Allogenic stem-cell transplantation (AlloSCT) involves the collection of stem cells from a matching donor and their transplantation into the patient to suppress the disease and restore their immune system; thus, it is the only treatment option in MF/SS with curative intention. It is used as second-line treatment for advanced stages of MF and SS, preferably in young patients with low tumor burden and a high predictable risk of progression at the same time. Problems with AlloSCT are non-relapse mortality (NRM) and graft versus host disease (GvHD) [130]. NRM is considered to be mortality due to the transplantation procedure. Median NRM is 21.5% and 23.6% at 1 and 3 years, respectively, mostly occurring within the first 3 months. NRM risk is higher in cases of myeloablative conditioning (compared with reduced-intensity conditioning), failure of three or more systemic treatments, and unrelated donor. Mean OS is 66% and 54% at 1 and 3 years, respectively. The same conditions that worsen NRM also reduce OS. GvHD appeared in 40% of cases but was severe in only 5.8% of them. Notably, GvHD is frequently provoked (reducing immune suppression) to enhance the graft vs. lymphoma reaction.

Providing patients with advice is very difficult. On the one hand, stage-based actuarial survival curves are not a guide to individual patient survival; on the other hand, NRM is around 20% in the first 3 months. The prognostic index (see MF/SS prognosis, above), especially PROCLIPI, will probably be helpful in the future for informing decisions.

In recent years, haploidentical bone marrow transplantation has also been performed on MF/SS patients [131], although broader series are needed to evaluate the efficacy and safety of this procedure compared with classical AlloSCT.

### 4.3. Targeted Therapies

The lack of curative treatment options for MF/SS means that many novel targeted therapies are currently being researched.

#### 4.3.1. Histone Deacetylase (HDAC) Inhibitors

In the last few years, a number of therapeutic agents, such as vorinostat and romidepsin (histone deacetylase inhibitors; HDACi), have been approved in the US and Japan, although no HDACi has so far been approved for use in Europe. HDACs are a class of enzymes that catalyze the removal of acetyl groups from histones, thereby restricting DNA accessibility and generally inducing epigenetic gene silencing. Although the mechanistic effects of HDACi in CTCL are not fully understood, they appear to induce upregulation of proapoptotic genes, alterations in the assembly of the kinetochore, and DNA damage (reviewed in [132]). Clinical trials have reported ORRs close to 30% for vorinostat and 34% for romidepsin, with 6% CRs for the latter, which seems to be more effective in SS [133,134,135]. A clinical trial for the treatment of MF with resminostat, another HDACi, as maintenance after response with previous treatments (RESMAIN, EudraCT 2016-000807-99, NCT02953301) is currently underway.

#### 4.3.2. Monoclonal Antibodies

In 2017 and 2018, two monoclonal antibodies—brentuximab vedotin and mogamulizumab—were approved in Europe after showing clinical benefits for CTCL patients.

Brentuximab vedotin (BV) is an antibody–drug conjugate composed of monomethyl auristatin E (MMAE), a cytotoxic anti-tubulin agent, and a chimeric monoclonal anti-CD30 antibody. It specifically binds to CD30 and, once internalized and cleaved, releases the MMAE that disrupts the microtubule network and causes cell-cycle arrest and apoptosis [136]. CD30 (*TNFRSF8*) is a cell membrane protein from the tumor necrosis factor receptor superfamily, which is widely expressed in MF/SS tumor cells [137]. In general terms, CD30 expression can change in very dynamically, increasing alongside disease progression, with most of the transformed MF cells exhibiting strong CD30 expression. In the ALCANZA study, an international randomized phase 3 trial, the median PFS was 15.9 months (versus 3.5 months for patients treated with the physician’s choice agent, bexarotene or methotrexate). A response duration longer than 4 months was achieved in 50% of the MF patients treated with BV, compared with 10% of those treated according to the physician’s choice, thereby confirming that this treatment is an appropriate option in patients with malignant cells expressing CD30 in at least 10% of the skin infiltrates. Although the results showed a weak positive correlation between levels of CD30 expression and BV sensitivity, some cases with lower intensity also appeared to benefit from the use of BV [138]. CRs of lesions with 0% positivity to CD30 expression have been noted, suggesting that expression of CD30 molecules on the cell surface at a level below which the IHC technique was not sufficiently sensitive to detect it could be enough to allow drug internalization and cell death [139].

Mogamulizumab is a humanized immunoglobulin G1 (IgG1) monoclonal κ antibody with a defucosylated Fc region that selectively binds to CCR4 (CCR4i). It exerts an enhanced antibody-dependent cellular cytotoxicity in CCR4-expressing tumoral cells [140]. CCR4 is expressed in almost all CTCL cases, albeit showing striking differences in the intensity and proportion of positive cells [141,142]. Gene expression studies have shown that CTCL cells express CCR4 at higher levels when compared with reactive lymphoid infiltrates [46]. It is also expressed in regulatory T cells (Tregs), which explains some of the side effects of mogamulizumab [143]. Deregulated CCR4 activity in CTCLs can be explained by increased expression of CCR4 and/or its ligands (CCL17 and CCL22) (see Figure 1), as well as by the acquisition of gain of function CCR4 mutations [26]. It is possible, that the genomic features described above might impact clinical responses to CCR4 blockage. In this regard, lessons learnt from adult t-cell leukemia/lymphoma (ATL) display improved responses to therapy using CCR4i, in those cases with mutated CCR4 [144].

In the MAVORIC study, CCR4i yielded a median PFS of 7.7 months (compared with 3.1 months with vorinostat) and ORRs of 21% in MF and 37% in SS patients with relapsed or refractory disease. Blood responses were higher (68% vs. 42% of skin responses) and faster (median time to response of 1.1 vs. 3.0 months in the skin) [142]. Interestingly, downstream of CCR4, a number of effector proteins can participate as part of its malignant activities (Figure 1). Thus, mutations in genes such as *NRAS*, *MEK1*, or *RHOA* have been detected in MF/SS cases and, therefore, may influence clinical sensitivity to CCR4i.

Other antibodies are currently being investigated for treating MF/SS patients, such as pembrolizumab (an immune checkpoint inhibitor that blocks PD-1) and lacutamab (IPH4102; an anti-KIR3DL2 monoclonal antibody).

Programmed cell death protein 1 (PD-1) is a cell-surface receptor expressed in activated T cells that transduces a negative signal-suppressing T-cell functions. The PD-1 gene (*PDCD1*) has been shown to be a haploinsufficient tumor suppressor that is frequently altered in T-cell lymphomas, and it is mutationally truncated, as well as heterozygously and biallelically deleted, in advanced MF and in SS patients (10–40% of cases) [26,27,32,38,43,145]. In addition, PD-1 expression has been detected in SS and some non-leukemic MF cases, frequently in the absence of other T follicular helper cell markers [146]. On the other hand, PD-L1 expression is basically restricted to macrophages and dendritic cells in CTCL samples. Under these conditions, clinical responses in advanced-stage MF/SS cases treated with pembrolizumab (ORR of 38%) were not correlated with the level of expression of PD-L1, the mutational burden, or an interferon-γ gene expression signature [147]. An EORTC clinical trial with atezolizumab (anti PD-L1 monoclonal antibody) is currently underway (EudraCT 2017-003680-35, NCT03357224).

KIR3DL2 (also known as CD158k) is a member of the killer-cell immunoglobulin-like receptor family that binds to HLA-class I ligands and negatively modulates immune cell functions. This receptor is widely expressed in malignant T cells in advanced MF/SS; hence, its inhibition could be a promising targeted therapy for these patients. In this regard, preliminary results from a phase I trial have shown an ORR in 44% of patients treated with lacutamab [148,149]. The TELLOMAK clinical trial is currently recruiting patients (NCT03902184).

#### 4.3.3. Other Targeted Inhibitors

Other clinical trials assessing the efficacy of targeted inhibitors for the PI3K, mTOR, CaN, and JAK/STAT pathways are currently being conducted.

Duvelisib and everolimus are oral inhibitors of PI3K and mTOR, respectively. These are being studied for the treatment of CTCL patients and have already shown partial responses with ORRs of 32% for duvelisib and 43% for everolimus [150,151].

Topical pimecrolimus, a CaN inhibitor, has been approved for treating several dermatological disorders, such as atopic dermatitis [152]. According to the accumulated data that identifies PLCγ1-downstream signaling as a major malignant mechanism in MF/SS, the clinical efficacy of targeting CaN using topical pimecrolimus is currently being addressed in a phase II clinical trial with patients at early stages (IA-IIA) of MF (PimTo-MF study, EudraCT 2014-001377-14).

Specific JAK inhibitors, such as ruxolitinib and tofacitinib, are being used systemically to treat myeloproliferative disorders that harbor an activated JAK2 mutant, as well as autoimmune diseases, such as rheumatoid arthritis, which has deregulated JAK/STAT activity [153,154], and they are used topically for cutaneous diseases such as alopecia areata and vitiligo [155,156]. Active clinical trials using ruxolitinib or tofacitinib in relapsed T or NK cell lymphomas, but not in CTCLs, are being conducted. STAT3 activation, which is associated with increased expression of the nuclear phosphorylated STAT3 form, has been described as a useful marker of MF cell transformation and clinical progression [42], making it an attractive therapeutic target for the treatment of this disease. This aim may be achieved by (i) inhibiting phosphorylation, (ii) preventing dimerization through the SH2 domains, (iii) blocking binding to the DNA, and/or (iv) downregulating expression. On the basis of the latter approach, AZD9150, a STAT3 antisense oligonucleotide, was evaluated in a phase Ib clinical trial in patients with diffuse large B-cell lymphoma in which STAT3 was associated with an aggressive disease phenotype. The study reported two CRs with a median duration of response of 10.7 months and two partial responses [157]. Although experimental and clinical findings support the use of JAK or STAT inhibitors in the treatment of CTCL, there is no formal proof that the clinical benefit depends on the pathway being activated.

## 5. Conclusions

The substrate for the clinical variability of MF/SS patients is dependent on the integration of a series of heterogeneous molecular alterations involving the TCR–PLCG1, CCR4/7–MAPK, TNFR/NF-κB, and JAK/STAT signaling pathways, with proliferative and survival signals derived from the MF/SS microenvironment. A more precise integration of all these variables can provide key prognostic and predictor information of value to the clinical management of MF/SS.

Weaknesses regarding MF/SS survival seem to be dependence on activated STATs, deregulated PLCG1 downstream activity, epigenetic control of transcription through HDACs, and the addiction of the neoplastic cells to the pro-tumorigenic signals from the microenvironment. These molecules offer robust rationales for developing novel therapeutic approaches. In this regard, very promising results have been obtained in clinical trials designed to target specific molecules such as CD30, KIR3DL2, and CCR4.

We may hope that clinical trials in development or being designed will increase the number of molecular targets, facilitate the use of combinations of multiple drugs, and potentiate the therapeutic stratification of patients through integrative genomic analyses.

## Figures and Tables

**Figure 1 cancers-13-01931-f001:**
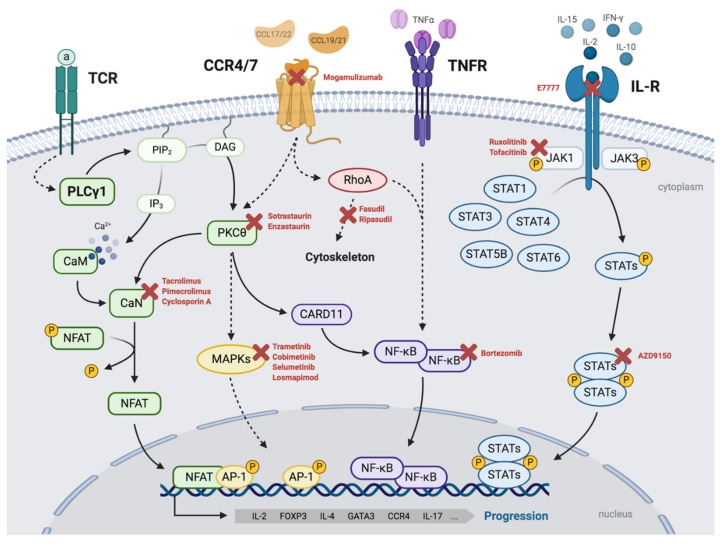
Malignant network of signaling mechanisms driving MF/SS. Malignant T-cell signaling pathways frequently deregulated in MF/SS cases that can drive the pathogenesis and progression of the disease and its potential targeted inhibitors. TCR/PLCγ1–NFAT/AP-1: upon stimulation with the antigen (a), TCR activates PLCγ1, which catalyzes the formation of IP3 and DAG. IP3 promotes Ca^2+^ release, which activates calmodulin (CaM), which, in turn, activates calcineurin (CaN). CaN dephosphorylates and activates the transcription factor NFAT, which can associate with the transcription factor complex AP-1 to induce gene expression. DAG activates PKCs (including PKCθ), which, in turn, induces the activation of MAPKs (and the AP-1 complex) and NF-κB signaling pathways. CCR4/CCR7-MAPKs/NF-κB: CCL17/22 and CCL19/21 activate CCR4 and CCR7, respectively, which can trigger the activation of downstream effectors such as MAPK–ERK, mediating gene transcription via AP-1 and NF-κB, and RhoA, which regulates cytoskeletal dynamics. TNFR–NF-κB: NF-κB can also be activated downstream of membrane receptors such as TNFR. IL-R/JAK–STAT: JAK–STAT signaling pathway can be activated downstream of transmembrane receptors such as interleukin receptors. JAK kinases are constitutively associated with the intracellular tails of the receptor. Upon activation of the receptor, JAKs are tyrosine-phosphorylated, making them catalytically active and capable of recruiting members of the STAT family and phosphorylating them at tyrosine residues. Upon phosphorylation, STATs form dimers that translocate to the nucleus, where they act as transcription factors for inducing gene expression. All of these pathways converge in inducing the expression of genes involved in the pathophysiology of malignant T-cells, such as IL-2, IL-4, L-17, FOXP3, GATA3, and CCR4. Created with BioRender.com (accessed on 18 November 2020).

**Table 1 cancers-13-01931-t001:** International Society of Cutaneous Lymphomas (ISCL)/European Organization for Research and Treatment of Cancer (EORTC) revision to the staging of mycosis fungoides (MF) and Sézary syndrome (SS). T: skin, N: lymph node, M: viscera, B: blood, OS: overall survival, DSS: disease-free survival. N/S: not specified. ^a^ Data extracted from [18].

Stage		T	N	M	B	Median OS (Years)	5 Year OS ^a^ (%)	5 Year DSS (%)
Early	IA	1	0	0	0–1			
		Limited patches or plaques <10% skin surface	No nodal involvement	No visceral involvement	<1000 atypical cells	35.5	N/S	98
	IB	2	0	0	0–1			
		Patches or plaques ≥10% skin surface	No nodal involvement	No visceral involvement	<1000 atypical cells	21.5	86	89
	IIA	1–2	1–2	0	0–1			
		Any patches or plaques	Aggregates of atypical cells	No visceral involvement	<1000 atypical cells	15.8	N/S	89
Advanced	IIB	3	0–2	0	0–1			
		Tumoral lesions	No involvement or aggregates of atypical cells	No visceral involvement	<1000 atypical cells	4.7	62	56
	III	4	0–2	0	0–1			
		Erythroderma	No involvement or aggregates of atypical cells	No visceral involvement	<1000 atypical cells	4.7	N/S	56
	IIIA	4	0–2	0	0			
		Erythroderma	No involvement or aggregates of atypical cells	No visceral involvement	<250 atypical cells	4.7	60	54
	IIIB	4	0–2	0	1			
		Erythroderma	No involvement or aggregates of atypical cells	No visceral involvement	250–1000 atypical cells	3.4	54	48
	IVA_1_	1–4	0–2	0	2			
		Any skin involvement	No involvement or aggregates of atypical cells	No visceral involvement	>1000 atypical cells + clonality	3.8	52	41
	IVA_2_	1–4	3	0	0–2			
		Any skin involvement	Partial or complete effacement of nodal architecture	No visceral involvement	<1000 or >1000 atypical cells + clonality	2.1	34	23
	IVB	1–4	0–3	1	0–2			
		Any skin involvement	Any nodal involvement	Visceral involvement	<1000 or >1000 atypical cells + clonality	1.4	23	18

**Table 2 cancers-13-01931-t002:** Next-generation sequencing studies in MF/SS.

Authors	Ref.	Number of Samples		WGS	WES	TS	RNA-Seq	scRNA-Seq	Highlighted Contribution
		MF	SS						
Lee et al., 2012	[45]	24	3				27		SS-associated lncRNAs
Vaqué et al., 2014	[32]	45	8			53			PLCG1, JAK mutants, and NFAT activation (IHC)
Sekulic et al., 2015	[33]		1	1			1		CTLA4:CD28 gene fusion
McGirt et al., 2015	[25]	30		5		25			JAK mut. and JAK inhibitors; C > T
Ungewickell et al., 2015	[34]	41	32		11	73			TNFR2 mut. and recurrent CTLA4:CD28 gene fusion
Choi et al., 2015	[27]		40	2	40				CNVs as drivers (STAT5B and PRKCQ amplification); C > T
Kiel et al., 2015	[35]		66	6	66				JAK1, 3/STAT3, 5B and ARID1A mut.
Pérez et al., 2015	[36]					35			JAK mut. and JAK inhibitors
Da Silva Almeida et al., 2015	[37]	8	25		42				Mutational landscape: CARD11 and cGKIβ mut.
Wang et al., 2015	[26]		37		37	37	32		TCR signaling and IL32, IL2RG expression; C > T
Prasad et al., 2016	[28]		12		12		10		ITPR1, 2, PKHD1L1 and DSC1 mut. and fusion genes; C > T
Woollard et al., 2016	[38]		101		10	101			POT1 mut., BRCA2 del. and PRKCQ, STAT3/5B amp; C > T
Izykowska et al., 2017	[39]		9	9					TOX and MYC amp. and deregulated expression
Litvinov et al., 2017	[46]	181					181 ^a^		Single-cell heterogeneity and transcriptional signatures for prognosis
Park et al., 2017	[40]	220 ^b^							RLTPR (NF-κB), CSNK1A1 and RHOA mut.
Chang et al., 2018	[44]	18 ^c^	121 ^c^						Mutually exclusive mut. within the NF-κB pathway
Bastidas Torres et al., 2018	[40]	9		9			8		HNRNPK and SOCS1 del. (JAK/STAT pathway)
Buus et al., 2018	[47]		11					11	Single-cell heterogeneity and surface marker expression
Perez et al., 2019	[41]	95				95			P-STAT3 (IHC) in advanced vs. initial MFs
Borcherding et al., 2019	[48]		1					1	Intra-patient heterogeneity
Gaydosik et al., 2019	[49]	5						5	Patient-specific heterogeneity and clustering
Iyer et al., 2019	[50]	27			10 ^d^		27		Clonotypic heterogeneity
Iyer et al., 2020	[43]	31			31				Divergent evolution of cancer subclones

MF: mycosis fungoides; SS: Sézary syndrome; WGS: whole-genome sequencing; WES: whole-exome sequencing; TS: targeted sequencing; scRNA-seq: single cell RNA-sequencing; lncRNAs: long non-coding RNAs; IHC: immunohistochemistry; C > T: transitions associated with ultraviolet (UV) mutational signature; Mut.: mutant; Del.: deletion; CNV: copy number variation; Amp.: amplification. ^a^ TruSeq targeted RNA expression assay; ^b^ data reanalysis from [25,26,27,28,32,34,35,37,38]; ^c^ data reanalysis from [25,26,27,28,34,37,38]; ^d^ probe-capture WES.

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
