# Peer review of "Mycosis Fungoides and Sézary Syndrome: An Integrative Review of the Pathophysiology, Molecular Drivers, and Targeted Therapy"

_cancers, 2021, doi:10.3390/cancers13081931_

Round 1
Reviewer 1 Report
Major points to be raised include:
- Authors must explain reasons for this increasing incidence (diagnostic methods, environmental factors?...). Page 2, paragraph 1.
- ‘Interestingly, the evolution of clàssic MF from patch, plaque to tumor and erythroderma develops in only 70% of cases’
This sentece is not correct. Not necessarily there is an evolution from one clinical lesion to another to advance in clinical stages.
Does occur in 70% of cases????
- ‘Histopathological examination frequently produces non-specific results. Infiltrate is
surprisingly scarce despite the intensity of skin lesions. Immunohistochemistry (IHC)shows positivity for CD3 and usually CD4, whereas CD7 and CD26 are usually negative.
Atypical cells express cutaneous lymphocyte-associated antigen (CLA, explaining the
skin-homing of Sézary cells), CCR4 and CCR7 (explaining the ability to spread beyond
the skin and to involve lymph nodes).’
CD26 is a flow cytometry marker. Cannot be perormed in skin biopsy specimens. CLA, CCR4, CCR7 are usually flow cytometry markers rather than histopathological characterization.
A definition of peripheral blood involvement in SS is lacking at this point.
- ‘These subunits are associated with the invariable subunits of the CD3 complex (γ, δ, ε and ζ) that enable signal transduction.’
The mean of the sentence seems not to be clear.
- ‘Gene expression studies have shown that CTCL cells continue expressing
CCR4 at a higher when compared with reactive lymphoid infiltrates.’
Check the mean of the sentence.
- ‘It is also possible that mutations affecting NRAS, MEK1 or RHOA, may influence
the clinical responses to this therapy.’
Authors may discuss the role/mean of mutations/genomic alterations in CCR4.
- Some additional typos or minor changes are highlighted and commented in the text.
Author Response
Dear Reviewer, thank you very much for your time and your comments with respect to this manuscript. Please see attachment. We include a full report that includes all the actions taken to response the commnets raised by the editor and the referees.
Sincerely,
J. Pedro Vaqué

Reviewer 2 Report
Congratulations to this thourough review! I believe that your work has the potential to become a milestone publication in the field. However, it would benefit from some updates, re-organization and shortening.
-Title: Too general an unspecific. Not really eye-catching. Your work is not really about diagnosis. The words "Pathophysiology" and "Review" should appear in the title.
-Simple summary: NGS is not defined. Here a short summary seems to be more adequate (e.g. ... here we review the classification, current knowledge about pathophysiology, treatment targets and upcomming treatment options)
-Abstract: Well written.
-Introduction: The first paragraph needs to be updated with current references on epidemiology... Consider replacing "dismal". Consider shortening paragraphs 1.1-1.3. Consider adding a section on large cell transformation, e.g. between 1.4-1.5.
-Table 1: Very clear and easy to read. Consider adding a thicker line between Early/Advanced. Typo in Stage IIA. Is the survival data relying on Mourad and Gniadecki 2020 JID?
-Table 2: Clear and easy to read. Consider adding a new heading on page 6 for easier readability. Consider adding a new column for single cell sequencing.
-Page 7: Typo: de Masson, not Masson.
-Page 13: 4.2.2: Consider reformulating last sentence more optimistic; e.g. Unfortunatly (...) IFN commercialisation was discontnued. PEG-IFN is currently the only available option, we strongly hope it´s manufacturing will continue since there is a urgent clinical need. (or similar)
-Page 13-14: Consider changing order of 4.2.3 and 4.2.4 (early stage-advanced stage treatment)
-Page 14: Typo gemcytabine
-Section 4.3: Why are drug names fat here and not in previous sections? Consider synchronyzing. HDACi, Anti-CD30 and Anti-CCR4 are not really new... Consider restructuring.
-References: Please consider an important shortening and updating, especially on epidemiology and survival and other recent reviews.
Author Response
Dear Reviewer 2, thank you very much for your time and your comments with respect to this manuscript. Please see attachment. We include a full report that includes all the actions taken to response the commnets raised by the editor and the referees.
Sincerely,
J. Pedro Vaqué
